# Interactions and pattern formation in a macroscopic magnetocapillary SALR system of mermaid cereal

Alireza Hooshanginejad[1], Jack-William Barotta [1], Victoria Spradlin[1,2], Giuseppe Pucci [3,4], Robert Hunt[1] & Daniel M. Harris [1] ✉

When particles are deposited at a fluid interface they tend to aggregate by capillary attraction to minimize the overall potential energy of the system. In this work, we embed floating millimetric disks with permanent magnets to introduce a competing repulsion effect and study their pattern formation in equilibrium. The pairwise energy landscape of two disks is described by a short-range attraction and long-range repulsion (SALR) interaction potential, previously documented in a number of microscopic condensed matter systems. Such competing interactions enable a variety of pairwise equilibrium states, including the possibility of a local minimum energy corresponding to a finite disk spacing. Two-dimensional (2D) experiments and simulations in confined geometries demonstrate that as the areal packing fraction is increased, the dilute repulsion-dominated lattice state becomes unstable to the spontaneous formation of localized clusters, which eventually merge into a system-spanning striped pattern. Finally, we demonstrate that the equilibrium pattern can be externally manipulated by the application of a supplemental vertical magnetic force that remotely enhances the effective capillary attraction.

Self-assembly is the spontaneous formation of organized structures or patterns that are governed by interactions between the individual constituents of a system[1]. Methods for controlling self-assembly at the microscale often focus on tuning the interactions between the constituents of the system through internal[2–4] or external means[5–7]. Mathematically, these interactions can be described by a pair interaction potential, which represents the work required to bring a pair of isolated particles from infinite separation to a finite distance apart[8]. In many cases, interaction potentials are composed of competing attractive and repulsive forces, with their relative strength and extent defining the possible structures achievable in the self-assembly of soft materials[9]. One class of competing interactions is defined by short-range attraction and long-range repulsion (SALR) forces, and has been documented at the microscale for proteins in low-salinity solutions[10–12]

and charged or magnetic colloidal particles in suspension[13–15], among others. Dense nuclear matter also represents an SALR system at the subatomic scale, with long-range Coulomb repulsion between protons competing with short-range attraction from the strong nuclear force mediated by neutrons[16]. Due to the attractive head and repulsive tail constituting SALR potentials energy landscape, they are sometimes referred to as mermaid potentials[17]. Overall, laboratory investigations of SALR systems are typically reserved to the microscale. In the present work, we realize a complementary macroscopic 2D SALR system that can be used to probe the fundamentals of equilibrium self-assembly in an accessible tabletop platform.

In general, short-range attraction favors particle clustering while long-range repulsion inhibits gelation[18–20], and large systems of particles often arrange into liquid-like states composed of finite equilibrium

[1]Center for Fluid Mechanics, School of Engineering, Brown University, Providence, RI, USA. [2]The Wheeler School, Providence, RI, USA. [3]Consiglio Nazionale delle Ricerche - Istituto di Nanotecnologia (CNR-NANOTEC), Via P. Bucci 33C, 87036 Rende, Italy. [4]Université Rennes, CNRS, IPR (Institut de Physique de Rennes) UMR 6251, FR35000 Rennes, France. ✉e-mail: daniel_harris3@brown.edu

clusters[19,21,22]. The patterns observed for such microscopic systems of particles primarily depend on the particle packing fraction and system temperature[22-24]. Regardless of the physical origin of such competing forces in a system of SALR particles, different structures can be predicted using models that apply techniques such as molecular dynamics[25-27] or Monte Carlo methods[28-30]. However, direct comparison between experiment and theory is frequently challenging, due to the subtle and often coupled physics involved in microscale realizations[17]. Nevertheless, while the precise mathematical form of the competing interactions inevitably differs between physical systems, the gamut of emergent patterns formed appears rather universal[31,32].

While commonly used to characterize interactions at the microscale, pairwise interaction potentials are relevant at all scales—for instance, the interaction of celestial bodies can be described by a pairwise gravitational potential. Of relevance to the present study, two bodies resting at a mutual fluid interface can be described by an interaction potential considering the gravitational and surface energy of the system. Identical axisymmetric particles tend to attract one another to minimize the total energy anomaly introduced by their presence, an effect often referred to as capillary attraction[33] or the Cheerios effect[34]. This phenomenon has been theoretically rationalized in various studies[33-40], and experimentally measured at both the microscopic[41-43] and macroscopic scales[44,45]. The capillary attraction between floating objects is a fundamental problem with relevance to applications involving self-assembly at fluid interfaces in natural systems[46-49] and laboratory settings[50-53]. Remote control and actuation of capillary assemblies is also a topic of recent interest, realized, for instance, through the external control of capillary interactions of deformable objects[54,55], the application of time-dependent external fields to which the particles respond dynamically[56], or the introduction of new physical effects such as thermocapillarity[57]. To prevent complete particle aggregation under capillary attraction, prior works have introduced repulsive magnetic dipole-dipole interactions through the application of an external magnetic field[56,58-60]. However, previous work at the macroscale has predominantly focused on short-range effects, with limited attention to the consequences of the relatively long-range nature of the magnetic repulsion[59]. Furthermore, the region of approximately uniform magnetic field in the Helmholtz coil geometry used in the prior experiments sets practical limits on the number of particles and the size of the domain that can be considered.

In this work, floating millimetric disks are embedded with permanent magnetic dipoles giving rise to a tunable SALR interaction potential. We isolate the possible pairwise equilibrium states in this macroscopic magnetocapillary SALR system through a combination of experiment and mathematical modeling, with the full landscape captured by two nondimensional parameters. After establishing the pairwise interaction landscape, we investigate the pattern formation of our magnetocapillary disks experimentally and numerically under varying packing fractions. We finally demonstrate external control of the self-assembly process in our magnetocapillary system by adjusting the capillary attraction strength remotely.

## Results

We systematically investigate the pairwise energy potential defined by capillary attraction and magnetic repulsion. We cast millimetric silicone-based hydrophobic disks of radius $a$ and mass $m$, and embed each with a vertically polarized cylindrical magnet of magnetic moment $M$ in its center. We gently deposit a magnetocapillary disk on the surface of a water-glycerol mixture followed by a second disk with a distance $l$ from the first disk, as shown in Fig. 1a. A complete description of the fabrication process for the magnetocapillary disks and fluid details are provided in the Methods section. Because the disks are hydrophobic, each disk rests in equilibrium on the fluid surface under capillary flotation[61] with the contact line pinned along the sharp bottom edge. In the absence of magnetic effects, two disks are fully attracted to each other under the Cheerios effect[34]. Although an exact analytical formula is not available for the capillary attraction force between finite disks, it has been shown through experiments, simulation, and scaling that the capillary attraction force between them can be well approximated by a decaying exponential of the form[45]

$$\mathbf{F}_c = -F_0\, e^{-l/\ell_c}\, \hat{e}_l. \qquad (1)$$

Here, $\hat{e}_l$ denotes the unit vector in the radial direction away from the adjacent disk, and $\ell_c = \sqrt{\gamma/\rho_m g}$ denotes the capillary length where $\gamma$ and $\rho_m$ denote water-glycerol surface tension and density, respectively. In this study, $\ell_c$ takes a fixed value of 2.4 mm for our experiments corresponding to the choice of working fluid. Also, $F_0$ is the characteristic attractive force found to be $2(mg)^2 a^{1/2}/\left(\pi^2 \gamma \ell_c^{3/2}[(a/\ell_c)^2 + 2a/\ell_c]^2\right)$ for circular disks at the capillary scale (i.e., $a \gtrsim l_c$)[45]. The exponential form is reminiscent of the shape of a 2D meniscus determined by the linearized Young-Laplace equation[62].

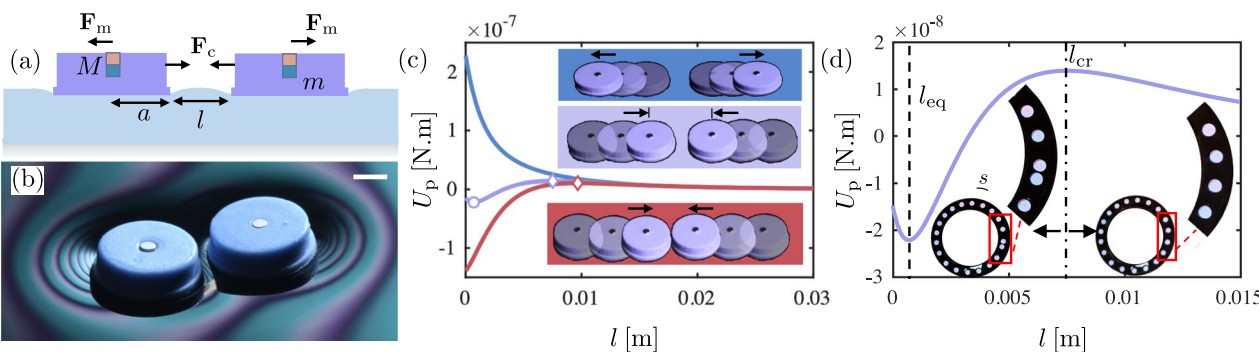

**Fig. 1 | Experimental setup and regimes of the magnetocapillary interaction potential. a** Schematics of two magnetocapillary disks of radius $a$ and mass $m$ each embedded with a small permanent magnet of magnetic dipole $M$. **b** In certain parameters regimes, two magnetocapillary disks find an equilibrium state defined by a finite spacing. The scale bar indicates 2 mm. **c** Sample magnetocapillary pairwise interaction potentials $U_p$ versus interdisk spacing $l$. The three characteristic regimes are shown here (corresponding to the sequel snapshots in the inset): the locally attractive or Cheerios (red) regime, the mermaid (violet) regime, and the fully repulsive (blue) regime. For all three cases $a = 3$ mm and $M = 9$ A·cm² with decreasing masses of $m_1 = 0.11$ g, $m_2 = 0.092$ g, and $m_3 = 0.080$ g, respectively. Note that if the disks are started sufficiently far apart, they will always repel, as the magnetic repulsion decays at a slower rate than the capillary attraction. The circle (○) and diamond (◇) markers indicate the locations of energy extrema for stable and unstable equilibria when $l > 0$, respectively. **d** Interaction potential for the sample mermaid regime. Inset: The experiment shows that when the mean spacing in an annular 1D array drops below $l_{cr}$, disks begin to spontaneously pair.

Since the magnet size is negligible compared to the disk center-to-center spacing $(2a + l)$, the magnets can be considered point dipoles with the repulsive magnetic force approximated as

$$\mathbf{F}_m = \frac{3\mu_0 M^2}{4\pi(l + 2a)^4} \hat{e}_l, \tag{2}$$

where $\mu_0$ denotes the permeability of free space[63]. The total pairwise interaction force is denoted by $\mathbf{F}_p = \mathbf{F}_m + \mathbf{F}_c$. Defining $U_m$ as the magnetic energy potential, and $U_c$ as the capillary energy potential where $\mathbf{F}_m = -\nabla U_m$ and $\mathbf{F}_c = -\nabla U_c$, the magnetocapillary interaction can also be expressed as an interaction potential $U_p = U_m + U_c$, defined as

$$U_p = \frac{\mu_0 M^2}{4\pi(l + 2a)^3} - F_0 \ell_c \, e^{-l/\ell_c}. \tag{3}$$

As evident in Eq. (3), $|\mathbf{F}_m|$ decays algebraically with $l$ while $|\mathbf{F}_c|$ decays exponentially. Depending on the input parameters, $\mathbf{F}_p$ can have 0, 1, or 2 roots (equilibria) for the physically relevant regime $l \geq 0$. As a result, three characteristically different interaction regimes emerge. If $\mathbf{F}_p(l = 0) < 0$ (i.e., locally attractive), then there is only 1 root, and it represents an unstable equilibrium at $l_{cr}$ where the interaction switches from attractive to repulsive, and is a consequence of the different decay properties of the two functions. Hence, the disks either repel each other for $l > l_{cr}$, or collapse into each other for $l < l_{cr}$. We refer to this regime as the Cheerios regime, since it features the local (complete) collapse behavior characteristic of the Cheerios effect[34]. Figure 1(c) shows $U_p$ as a function of $l$ for a sample Cheerios regime with the solid red line.

If $\mathbf{F}_p(l = 0) > 0$ (i.e., locally repulsive), $\mathbf{F}_p$ can have either 0, or 2 roots for $l > 0$. The former is referred to as the fully repulsive regime, and an example is represented by the solid blue line in Fig. 1c. When $\mathbf{F}_p$ has 2 roots, the first root, $l_{eq}$, is a stable equilibrium point, while the second root, $l_{cr}$, is an unstable equilibrium point with example as the purple solid lines in Fig. 1c, d. Under such a scenario, $l_{eq}$ represents a potential well, where the two disks will reach equilibrium at a finite spacing as shown experimentally in Fig. 1b, c inset. We refer to this regime as the mermaid regime since it presents a SALR system most reminiscent of the SALR systems previously realized at the colloidal scale[17]. Note that the Cheerios regime also formally represents a SALR regime, but the presence of static friction due to the possibility of physical contact between the disks renders the exploration of pattern formation less predictable and reproducible. In the mermaid regime, a pair of disks repel each other if they are positioned $l > l_{cr}$ apart, but fall into $l = l_{eq}$ if $l < l_{cr}$. The experimental realizations shown in Fig. 1c show that for a single disk size and magnetic strength, the three regimes can be accessed by gradually increasing the disk mass, therein increasing the strength of the capillary attraction alone, effectively moving from the fully repulsive to mermaid to Cheerios regimes. Videos showing pairwise particle interactions in the mermaid and fully repulsive regimes are provided in Supplementary Videos 1 and 2, respectively.

Figure 1d inset images show a segment of a 1D array of magnetocapillary mermaid disks confined in an annular channel of width 15 mm with the corresponding pairwise energy potential presented in Fig. 1d. Under low-packing fractions, the disks are all positioned above the $l_{cr}$ threshold, securely in the repulsive tail of the energy potential. Given the channel's central radius, $R$, and the number of magnetocapillary mermaid disks, $N$, the mean distance between the disks, $s$, can be estimated as $(2\pi R - 2Na)/N$. For the magnetocapillary mermaid regime demonstrated in Fig. 1d, $l_{cr} = 7.5$ mm. As shown in Fig. 1d inset, when $R = 6$ cm and $a = 6$ mm, for $N = 19$ the disks are spaced uniformly with anticipated $s = 7.8$ mm. However, by adding one more disk to the channel (i.e., $N = 20$), the anticipated spacing under uniform distribution is 6.8 mm, now lower than $l_{cr}$. Under this scenario, the new disk

pairs up with the closest disk at the equilibrium spacing position (Fig. 1d inset), breaking the symmetry of the original 1D crystalline pattern.

We fabricated disks with different radius ($a$), mass ($m$), and magnetic moment ($M$) to more exhaustively verify our magnetocapillary potential model, and identify the limits of the various interaction regimes. To facilitate this exploration, we first normalize $\mathbf{F}_p$ by the characteristic capillary force $F_0$, and the disk spacing $l$ by the disk diameter $2a$. Hence, we arrive at a nondimensional interaction force

$$F_p^* = \mathcal{M}(l^* + 1)^{-4} - e^{-2\sqrt{Bo}\,l^*}, \tag{4}$$

where the asterisks denote the nondimensionalized force and spacings. Two nondimensional parameters emerge that govern the interaction: the gravitational Bond number $Bo = (a/\ell_c)^2$ that represents the strength of gravity to capillary forces, and a magnetocapillary number $\mathcal{M} = 3\mu_0 M^2/64\pi a^4 F_0$ that represents the ratio between the magnetic and capillary forces, in a similar spirit to prior work[59] but adapted to our larger length scales.

We find the roots of $F_p^*$ for $l^* > 0$ and varying $Bo$ and $\mathcal{M}$. For $Bo \leq 4$ three possible states exist, which we delineate by the number of roots satisfying $l^* > 0$. The transition between 0 roots to 2 roots marks the boundary between the fully repulsive regime and the mermaid regime, and is demarcated by the curve $\mathcal{M} = \frac{16}{Bo^2} e^{-2\sqrt{Bo} - 4}$. The transition from 2 roots to 1 root marks the boundary between the mermaid and Cheerios regimes and is demarcated by $\mathcal{M} = 1$. These predictions are summarized in Fig. 2a by the dashed lines, showing that the mermaid regime is only possible in a finite range of parameter space. The color spectrum indicates the predicted equilibrium spacing distance $l_{eq}^*$ (when finite). At $Bo = 4$, $\mathcal{M} = 1$, a triple point is predicted to exist between the three phases, and above $Bo \geq 4$ the mermaid regime is predicted to vanish entirely. For $Bo \geq 4$, only the Cheerios and fully repulsive regimes exists, switching over at $\mathcal{M} = 1$.

The root corresponding to the inevitable transition to long-range repulsion ($l_{cr}^*$) can be found analytically, and is given by

$$l_{cr}^* = -\frac{2}{\sqrt{Bo}} \mathcal{W}_{-1}(\Theta(\mathcal{M}, Bo)) - 1. \tag{5}$$

Here, $\mathcal{W}_k$ represents the $k$-th branch of the Lambert $\mathcal{W}$ function, and its argument $\Theta(\mathcal{M}, Bo) = -\frac{1}{2}\sqrt{Bo}\,\mathcal{M}^{1/4}e^{-\sqrt{Bo}/2}$. As described above, $l_{cr}^*$ is a positive finite value in both the Cheerios and mermaid regimes. Furthermore, the second root ($l_{eq}^*$) can be expressed as

$$l_{eq}^* = -\frac{2}{\sqrt{Bo}} \mathcal{W}_0(\Theta(\mathcal{M}, Bo)) - 1, \tag{6}$$

and takes a finite positive value only in the mermaid regime. For $\mathcal{M} < 1$ (i.e., in the Cheerios regime) the value of ($l_{cr}^*$) becomes negative, corresponding to a physically inaccessible equilibrium location inside the disk. Steric contact forces between the disks thus balance the residual local attraction in this regime, with the disks in solid contact with one another. More details regarding exact solutions for the equilibrium points and phase boundaries can be found in Supplementary Note 3.

The overlaid points in Fig. 2a show the experimentally observed interaction regime under different combinations of parameters. The experimental points in the mermaid regime are similarly color-coded by their measured equilibrium spacing. Complete details of the parameters associated with each experimental symbol are presented in Supplementary Note 2. The model prediction for both the extent of the mermaid regime and the measured $l_{eq}$ value are in very good agreement with the experimental data, validating the accuracy of our magnetocapillary SALR interaction model. Further quantitative comparison of $l_{eq}$ for the controlled parameters is provided in Supplementary Note 1.

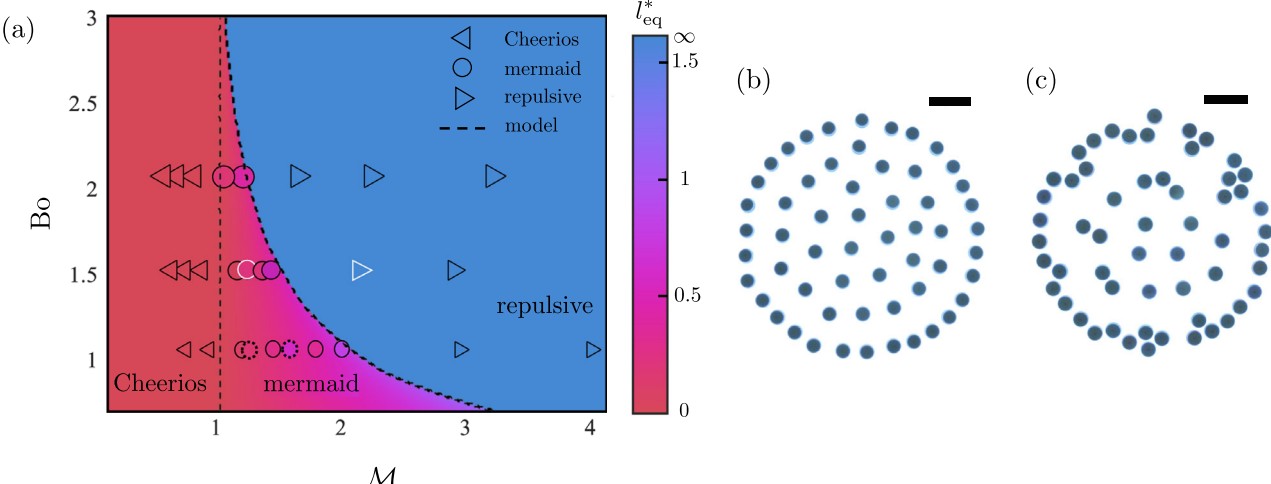

**Fig. 2 | Magnetocapillary interactions. a** The phase diagram for different regimes as a function of capillary Bond number Bo and magnetocapillary number $\mathcal{M}$. The left triangles (◁) show the Cheerios regime, the right triangles (▷) show the repulsive regime, and the circles (◯) show the mermaid regime, as observed in the experiments. The symbol sizes correspond to three disk sizes, $a = 2.5$ mm, $a = 3$ mm, and $a = 3.5$ mm, hence three different Bo in ascending order. The phase diagram is color-coded based on $l_{eq}^*$ computed from the model, while the circles are color-coded based on $l_{eq}^*$ value measured in the experiments. The symbols with solid outlines represent magnets with $M = 9$ A·cm² while the dashed-line outlined symbols represent disks with permanent magnets of $M = 2.25$ A·cm². While Bo is only varied by $a$ in our experiments, $\mathcal{M}$ is varied with $a$, $m$, and $M$. The dashed lines indicate the model prediction for boundaries between different regimes. The dimensional parameters and equilibrium spacing for each experiment is provided in Supplementary Note 2. **b** The equilibrium configuration in the experiment for fully repulsive disks with $\mathcal{M} = 2.1$, Bo $= 1.52$, and $\phi = 0.1425$ ($a = 3$ mm, $m = 0.073$ g, $M = 9$ A·cm⁻², $R = 6$ cm, $N = 57$), also marked with the white border right triangle in the phase diagram. **c** The equilibrium configuration in the experiments for mermaid disks with $\mathcal{M} = 1.22$, Bo $= 1.52$, and $\phi = 0.1425$ ($a = 3$ mm, $m = 0.092$ g, $M = 9$ A·cm⁻², $R = 6$ cm, $N = 57$), also marked with the white border circle in the phase diagram.

We perform 2D experiments to investigate how the pairwise interactions in magnetocapillary disks scale up in larger collections under the various regimes presented in Fig. 2a. A circular confinement corral is filled with a water-glycerol mixture, and has a 1.5 cm opening connected to a smaller bath where the disks are deposited and individually directed toward the main confinement. The pattern is allowed to stabilize before the next disk is introduced. The confinement corral is slightly underfilled, so the border meniscus acts as a non-steric repulsive barrier. We control the magnetocapillary disks areal packing fraction $\phi = Na^2/R^2$, where $N$ denotes the number of disks in a circular 2D confinement, and $R = 6$ cm denotes the confinement radius. In the absence of magnets, the disks collapse into a single granular raft under unimpeded capillary attraction[64]. As shown in Fig. 2b, fully repulsive disks with $a = 3$ mm, $m = 0.073$ g, and $M = 9$ A·cm⁻² ($\mathcal{M} = 2.1$ and Bo $= 1.52$) exhibit a hexagonal crystal lattice with defects at $\phi = 0.1425$. The hexagonal lattice has an inhomogeneous structure as a result of the finite confinement. For lesser confinement, the lattice structure is expected to become more regular. However, as shown in Fig. 2c mermaid disks of the same size and same magnet dipole strength with slightly higher mass at $m = 0.092$ g ($\mathcal{M} = 1.22$ and Bo $= 1.52$) form clusters at $\phi = 0.1425$. Due to confinement effects, disks first begin pairing along the boundary. Videos of the experimental self-assembly processes for both the repulsive and mermaid regimes are provided in Supplementary Videos 3 and 4, respectively. We use the pairwise interaction potential provided in Eq. (3) to perform an $N$-body simulation on a confined 2D system of $N$ magnetocapillary disks. In addition, a localized confining potential models the boundary of the fluid container. The overdamped equation of motion for each disk $i$ is

$$\frac{\mu(\pi a^2)}{H}\dot{\mathbf{x}}_i = \mathbf{F}_{\text{confining},i} + \sum_{j \neq i}\left(\mathbf{F}_{\text{c},ij} + \mathbf{F}_{\text{m},ij}\right), \qquad (7)$$

where $\mathbf{x}_i$ denotes the position vector for disk $i$ with respect to the center of the circular confinement, the dot represents the derivative with respect to time, and $H$ and $\mu$ are the liquid bath depth and viscosity, respectively. The confining force, $\mathbf{F}_{\text{confining}} = -\nabla U_{\text{confining}}$, is assumed to be governed by a confining potential of form $U_{\text{confining}}(r) = \frac{\alpha}{2}\left[1 + \tanh\left(\frac{r+a-R}{\ell_c}\right)\right]$. Although the exact functional form is not important here, the form used is chosen to provide a repulsive barrier that decays away from the system boundary over the capillary length scale, capturing the influence of the repulsive meniscus barrier in a fluid container. Here, $r$ denotes the radial distance from the disks center to the center of the circular confinement, $R$ denotes the confinement radius, and $\alpha$ is a constant corresponding to the depth of the potential well tuned to be sufficiently large to prevent particles from escaping the corral. The initial positions of the disks are sampled from a uniform distribution with resampling occurring in the case of disk-disk overlap. The positions of all disks are then evolved in time as $2N$ first-order ODEs according to Eq. (7). The ode45 solver in MATLAB is used to iterate the system of ordinary differential equations. The underdamped version of Eq. (7) (i.e., $4N$ equations) gives indistinguishable results for the equilibrium state, justifying the overdamped simplification for our particular exploration. Additional details on the simulation setup and implementation can be found in Methods.

To further study the equilibrium pattern formation of the magnetocapillary disks in the mermaid regime, we perform 2D experiments and simulations under different areal packing fractions $\phi$. As shown in Fig. 3a top row for $a \ll R$, the system exhibits a hexagonal lattice state at $\phi = 0.05$. As we increase $\phi$ in 0.05 increments, the magnetocapillary mermaid disks begin pairing spontaneously, with clusters forming at $\phi = 0.1$ and 0.15. At $\phi = 0.2$ the system shows stripe patterns until transitioning to labyrinths at $\phi = 0.25$. Figure 3a, bottom row, shows the simulation results under confinement for the same physical parameters as in the experiments while varying $\phi$, with good agreement. To explore any influence of geometrical confinement on the pattern formation (a practical necessity in the experiment), simulations are also completed with periodic boundary conditions in a large square domain (17 cm × 17 cm) that is 2.5 times larger than the confinement area in the experiments. As shown in Fig. 3b, similar patterns and transitions arise, albeit with more spatial homogeneity due to the absence of a true border.

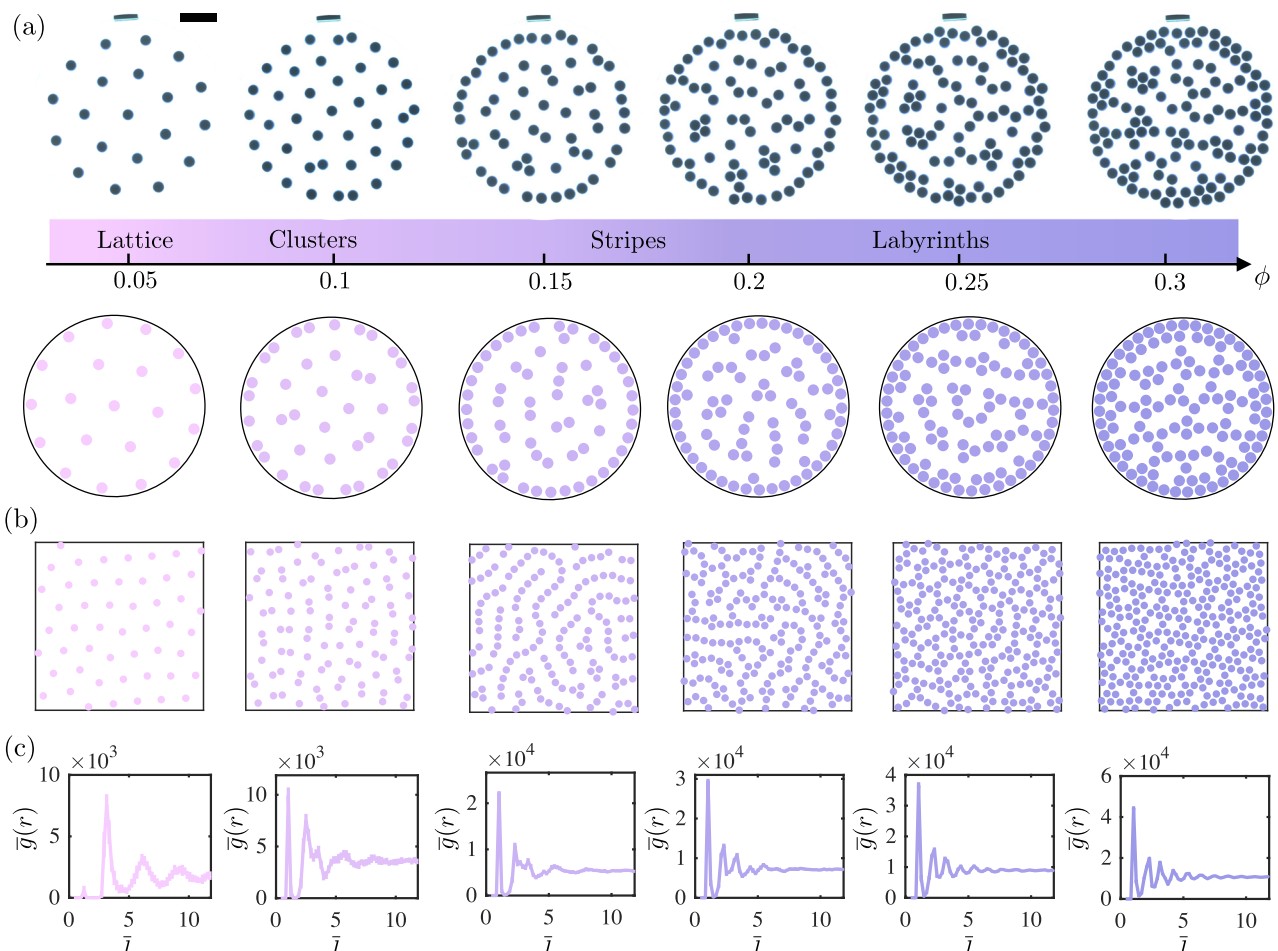

**Fig. 3 | 2D pattern formation in the magnetocapillary mermaid regime.**
**a** Patterns formed by magnetocapillary mermaid disks with $\mathcal{M} = 1.22$ and Bo = 1.52 ($a = 3$ mm, $M = 9$ A · cm$^{-2}$, $m = 0.092$ g) for varying areal packing fraction, $\phi$. The confinement radius $R$ is 6 cm. The top row illustrates the experimental snapshots starting from the hexagonal lattice state to clusters, stripes, and labyrinths. The bottom row shows sample model results color-coded based on corresponding $\phi$ values in the experiments. The black scale bar shows 2 cm. **b** Simulation results of

mermaid disks with the same parameters as (**a**) in a large square domain of 17 cm × 17 cm with a periodic boundary condition for varying packing fraction, $\phi$. **c** The average radial distribution function RDF, $\overline{g}(r)$, from the periodic boundary condition simulations in a square domain of 17 cm × 17 cm as a function of $\overline{l}$ averaged over 25 independent simulations for varying $\phi$. The error bars indicate one standard deviation of the simulation results.

To quantitatively characterize the simulated patterns in 2D, we construct a radial distribution function (RDF)[32]. The RDF for disk $i$, $g_i(r)$, is given by $g_i(r) = K_{i,[r, r + \Delta r]}/(2\pi r \Delta r)$ where $K_{i,[r, r + \Delta r]}$ denotes the number of disks located at a distance of $r$ to $r + \Delta r$ from disk $i$. We can then obtain the average RDF for the entire system as $g(r) = \frac{1}{N}\sum_i^N g_i(r)$. To characterize the system's phase under the mermaid potential, we average $g(r)$ over 25 independent simulations for each $\phi$ in a square domain of 17 cm × 17 cm with periodic boundary conditions. Figure 3c shows the average RDF $\overline{g}(r)$ as a function of the distance from the disks normalized by the equilibrium distance between two isolated disks, namely $\overline{l} = (l + 2a)/(l_{eq} + 2a)$. As indicated in Fig. 3c, $\overline{g}(r)$ is small at $\phi = 0.05$ for $\overline{l} < 2$, with a maximum at $\overline{l} \approx 3.1$ which is the characteristic spacing in the hexagonal lattice formation for this packing fraction. For $\phi = 0.1$, the strongest peak emerges at $\overline{l} \approx 1$ indicative of the magneto-capillary mermaid stable equilibrium state, which we identify as the transition to the cluster phase. With increasing $\phi$, the clusters coalesce into longer chains of disks, hence, additional peaks appear at integer multiples of $\overline{l} \approx 1$ indicative of the striped phase. The stripe and labyrinth phases exhibit similar RDFs[32], although the labyrinth phase has visual signatures of a more globally connected network of stripes. In comparison to the low-packing fraction hexagonal lattice state, the intermediate packing fractions display a more short-range order. As the packing fraction is further increased, it is anticipated that a long-

range order should be restored as a dense hexagonal lattice emerges, with particle-particle separation approximately $\overline{l} = 1$. These phases undergo a topological transition and could potentially be distinguished by alternative metrics, such as the corresponding Betti numbers or average pore size.

We further aim to showcase the tunability of our macroscopic system by manipulating the system phase remotely. We place a large magnetic sheet below the bath ~7 mm below the free surface as depicted in Fig. 4a. A pair of magnetocapillary disks with $a = 2.5$ mm, $m = 0.09$ g, and $M = 9$ A · cm$^2$ (Bo = 1.05, $\mathcal{M} = 3.5$) show fully repulsive behavior as shown in Fig. 4b inset. However, as we place the magnetic strip below the bath, the disks are pulled down by the applied vertical magnetic force between the embedded magnets and the magnetic sheet resulting in a larger depression of the meniscus surrounding the disks, and hence a larger capillary force between the disks as illustrated in Fig. 4a. This force increase is such that the fully repulsive disks in Fig. 4b transition to the mermaid regime and fall into the potential well configuration.

In the absence of a detailed characterization of the magnetic sheet, the increase in the capillary force is modeled by replacing the mass of the disk with an effective (greater) mass that yields the equilibrium spacing experimentally measured in a two-disk experiment under the additional uniform vertical magnetic force. From the

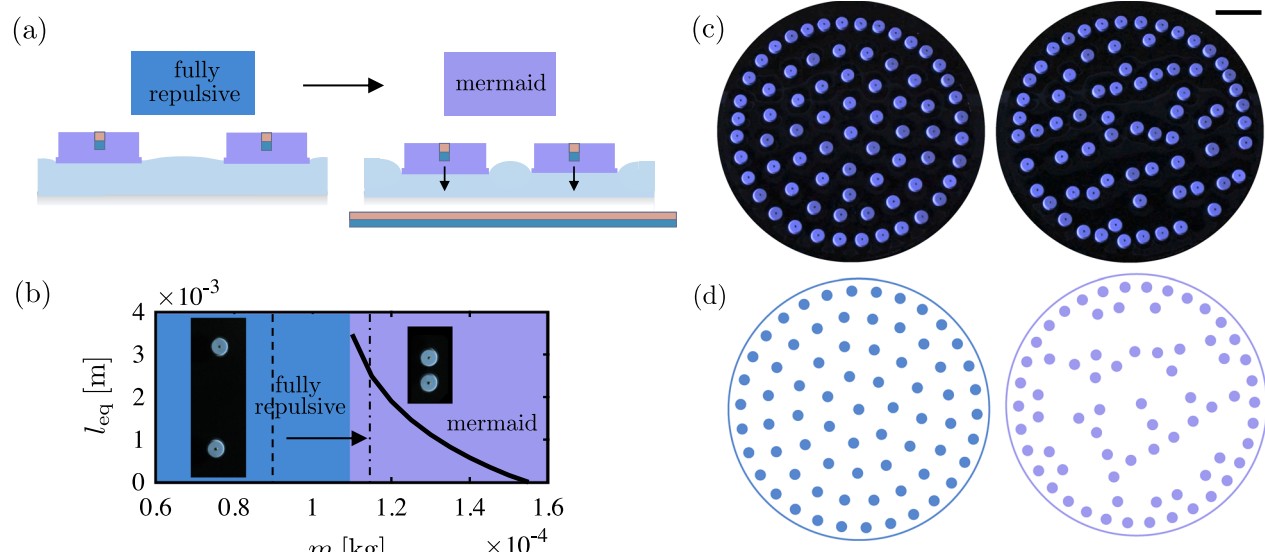

**Fig. 4 | External control of equilibrium pattern. a** When a magnetic strip is placed below the bath, the disks are pulled downwards, increasing the capillary attraction force. If tuned correctly, fully repulsive disks can transition to mermaid disks under this additional force. **b** $l_{eq}$ as a function of $m$ from the magnetocapillary model for $a = 2.5$ mm, and $M = 9$ A·cm². The dashed line shows the physical value of $m$ while the dotted dashed-line shows the effective $m$ after applying the external magnetic force based on the $l_{eq}$ value extracted from the experiments shown in the inset.

**c** The hexagonal lattice state (on the left) transitions to the striped pattern (right) by adding the magnetic strip below the bath for disks with $a = 2.5$ mm, $M = 9$ A·cm², and $m = 0.09$ g (Bo = 1.05, $\mathcal{M} = 3.5$) and $\phi = 0.18$. The scale bar indicates 2 cm. **d** Sample magnetocapillary simulations for $a = 2.5$ mm, and $M = 9$ A·cm² showing phase transition from hexagonal lattice to stripes when $m$ changes from 0.09 g to 0.115 g (Bo = 1.05 with $\mathcal{M}$ decreasing from 3.5 to 1.9) for $\phi = 0.18$.

experimental $l_{eq}$ under the magnetic strip, we find the effective mass that yields the same $l_{eq}$ using the model. Figure 4b shows the model solution with the solid line for $l_{eq}$ versus $m$ for $a = 2.5$ mm, and $M = 9$ A·cm². The dashed line shows the disks original mass (0.09 g), and the dotted dashed line shows the effective mass (0.115 g) corresponding to $l_{eq} = 2.6$ mm from the experimental inset. Therefore, we can model the system before and after adding the external magnetic strip by changing $m$ from 0.09 to 0.115 g in the magnetocapillary potential. Figure 4c, d show a 2D system of fully repulsive disks with the same disk parameters from experiments and simulations, respectively, for $\phi = 0.18$. The disks form a hexagonal lattice away from the boundary until the external magnetic strip is placed beneath, upon which the system immediately transitions to a stripe state. A video of this process is included as Supplementary Video 5. As shown in Fig. 4c, d, the simulation results with the effective mass is in good qualitative agreement with the experiments. While in our case the particles themselves do not deform under the additional applied field, other recent work has demonstrated more advanced control of capillary multipole interactions through magnetically responsive flexible floating objects[54].

## Discussion

In summary, we have demonstrated that the competing interactions between the attractive capillary force and the repulsive magnetic force can result in a macroscopic SALR potential with rich self-assembly phenomenology. The lattice, cluster, and stripe patterns are reminiscent of emergent patterns in various colloidal SALR systems[65–68]. Beyond colloids, these types of patterns also arise in certain hard condensed matter systems. Crystal transition to dimer crystal to charge stripe in melting Wigner crystals[69,70] and pasta phases in dense nuclear matter[16] are examples of hard condensed matter SALR systems with similar patterns. We have developed a magnetocapillary potential model defined by two nondimensional parameters that rationalizes the pairwise interaction between the magnetocapillary disks, and reproduces different phases of pattern formation for varying packing fractions. In addition, we have shown that by manipulating the

capillary force, we can control the patterns externally. The current magnetocapillary SALR system provides an accessible playground for investigating analogous aspects of SALR systems across colloidal scales to subatomic scales where experiments are more challenging or expensive. For instance, there are a number of recent theoretical and numerical predictions that lack experimental validation that may be good candidates for future work with our system, such as the formation of spiral patterns in confinement[24] or pattern formation over periodic substrates[71]. Additionally, the system could be fruitfully applied to pedagogy for introducing modern topics in self-assembly or allow researchers to develop a better physical intuition on analogous microscale systems. To further establish connections with colloidal-scale SALR systems, future studies will explore simulating the effect of thermal fluctuations and Brownian motion via the addition of super-critical chaotic Faraday waves[72,73].

## Methods
### Experimental details
The disks were fabricated using OOMOO 30 which is a silicone rubber compound comprised of part A and part B that should be mixed 1:1 in volume or 10:13 in weight, respectively[74]. The density of the mixed compound under this scenario was $1.34 \pm 0.02$ g·cc⁻¹. We mixed 30 ml of part A and 30 ml of part B for 30 s by hand with a wooden craft stick. We then poured the compound into a 3D-printed mold that was printed using a Formlabs Form 2 resin printer. After pouring the compound into the mold, the mold was placed in a small vacuum chamber and degassed for 5 min to extract air bubbles trapped inside the mold. Next, we scraped the surface of the mold using a blade to make a smooth and balanced surface on the mold. The compound cured inside the mold for at least 6 h in room temperature. We then used a compressed air nozzle to remove the disks from the mold.

In order to robustly achieve a smooth and axisymmetric three-phase contact line when depositing disks, we designed the molds with a thin lip around the disks edge that was 0.25-mm high and 0.25-mm thick. We also cast a cylindrical recession in the center of the disk for placing magnets. We used cylindrical Neodymium 50 magnets

(Supermagnetman) of 1 mm height and two different diameters of 0.5 mm and 1 mm (corresponding to $M = 2.25$ A · cm$^2$ and $M = 9$ A · cm$^2$, respectively). To place a magnet inside a disk, we used the south pole of a permanent magnet underneath the disks to get a magnet inside the hole. We then removed the permanent magnet while manually holding the installed magnet in the disk.

For the working fluid, we used a mixture of water-glycerol with density, $\rho_m = 1139.0 \pm 0.3$ kg · m$^{-3}$, measured with a density meter (Anton Paar DMA 35A), surface tension, $\gamma = 0.068 \pm 0.001$ N · m$^{-1}$, and dynamic viscosity, $\mu = 0.0083 \pm 0.0004$ N · s · m$^{-2}$ [75]. The fluid is gently poured into a circular confinement corral made from laser-cut acrylic sheets of thickness 6 mm to a depth of $H = 5$ mm.

## Simulation details

All simulations are carried out in MATLAB using the known physical parameters from the corresponding experiments as detailed in the figure captions. The `ode45` solver (explicit Runge-Kutta (4,5) formula) in MATLAB is used to iterate the system of ordinary differential equations for both the confined and periodic domains. All simulations begin by randomly placing $N$ disks across the domain, sampling from a uniform distribution. Resampling occurs in the case where two positions are sampled that would correspond to a physical overlap of two disks in the experiment. For the case of circular confinement, the sampling region of the domain is restricted to a circle of radius $R - \ell_c$ so that no disks are placed on the meniscus of the container of radius $R$. Eq. (7) was simulated for a sufficiently long time such that the energy of the final configuration was constant for an extended period, ensuring that a minimal energy configuration was attained.

For the periodic domain, we initialize $N$ particles inside a square domain of side length $L$. At each time step, we place an image square of disks to the left, right, above, below, and on all four of the diagonals, and compute the total net forces on the original particles. Due to the large size of the domain relative to the extent of the competing interaction forces (magnetic/capillary), no additional images are needed to mimic a periodic boundary. As such, each individual particle responds to forces from $9N - 1$ particles (i.e., $N - 1$ particles in the original domain, and $8N$ image particles). No additional confinement potential is used in the periodic simulations.

For the purpose of computing the RDFs from the outputs of the simulations, we set $\Delta r = \ell_{eq}$, which is a small but physically meaningful length scale in the problem.

## Data availability

The data that support the findings of this study are available from the corresponding author upon request.

## Code availability

All code and sample data associated with the numerical simulations can be found in ref. 76 and https://github.com/harrislab-brown/MermaidCereal.

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

## Acknowledgements

The authors would like to acknowledge Ryan Poling-Skutvik, Charles Reichhardt, and Cynthia Reichhardt for feedback and fruitful discussions. A.H. acknowledges the Hibbitt Fellowship. J.-W.B. is supported by the Department of Defense through the National Defense Science and Engineering Graduate (NDSEG) Fellowship Program. G.P. acknowledges the CNR-STM and the CNRS-Momentum Programs. This work is partially supported by the Office of Naval Research (ONR N00014-21-1-2816).

## Author contributions

A.H., J.-W.B., G.P., and D.M.H. designed research, analyzed data, developed models, and wrote the paper; A.H. and V.S. performed final experiments; A.H. and R.H. analyzed the models; J.-W.B. developed and performed simulations; all authors performed research, discussed the results, commented on the manuscript, and gave final approval for publication, agreeing to each be held accountable for the work performed therein; D.M.H. supervised the project and secured funding.

## Competing interests

The authors declare no competing interests.
