## [Peer Review File · Nature Communications]

REVIEWER COMMENTS

Reviewer #1 (Remarks to the Author):

Review for the manuscript Mermaid Cereal: Interactions and Pattern Formation in a Macroscopic Magnetocapillary SALR System by Alireza Hooshanginejad, Jack-William Barotta, Victoria Spradlin, Giuseppe Pucci,³ Robert Hunt, and Daniel M. Harris.

In this work, the authors used floating millimetric disks at a fluid interface, urged by a capillary attraction. The disks also have permanent magnets to introduce a competing repulsion effect. They study their pattern formation in equilibrium, where a short-range attraction and long-range repulsion interaction potential describe the pairwise energy landscape of two disks. They do experiments and simulations, demonstrating that the equilibrium pattern can be externally manipulated by applying a supplemental vertical magnetic force that enhances the effective capillary attraction.

In my opinion, this is an outstanding work. The data obtained through experiments sounds technically well-developed and exhaustively analyzed and interpreted. Also, I consider the author's conclusions consistent with their findings. In this manuscript, one can learn the physical consequences of modifying common potentials (repulsion at the short range and attraction at the long-range). Ideal for teaching statistical physics. As the authors wrote: The current magnetocapillary SALR system provides an accessible playground for investigating analogous aspects of SALR systems across colloidal scales to subatomic scales where experiments are more challenging or expensive, and could also be used in pedagogy when introducing modern topics in self-assembly. This manuscript must be published.

However, I have a few questions that have to be addressed by the authors:

- 1) In lines 196-198, they mention that fully repulsive diskshexibit a hexagonal crystal lattice. This surely occurs for large corrals, but they are only distorted hexagons here. Please comment on that.
- 2)Is the order long- or short-ranged for the average radial distribution function RDF, $g(r)$, for the first column in Fig 3?
- 3)In Methods, you must include details of the used simulation method.

Reviewer #3 (Remarks to the Author):

The authors describe an original experimental system, where floating objects embedded with permanent magnets present various interaction regimes. In the most complex regime, the long range repulsion generated by the magnets and the short range attraction induced by the Cheerios effect compete to create a "mermaid" potential, similar to microscopic systems. The most original observation of the authors resides in the apparition of various self-assembled phases due to this mermaid potential, similar to what is observed in microscopic scales. The authors determined the conditions in which the mermaid potential is obtained with analytical equations, and also provide numerical simulations matching closely their experiment.

The work of the authors is of very sound nature and they provide all required information to reproduce their experiments and simulations, including some open access version of their code, which I really appreciate. Their results are potentially interesting as they could open the way to mimic structures observed with microscopic systems.

Regarding the literature review, I think that the authors overlooked a few of the most recent publications involving controlled self-assembly of floating objects. Here are some examples from a lab I am familiar with:

DOI: 10.1039/d0sm01251c

DOI: 10.1039/d1sm00447f

DOI 10.1140/epje/i2017-11599-y

The last one (Metzmacher, Poty et al., 2017) in particular also studies the self-assembly of floating objects with embedded permanent magnets, whose capillary interaction is governed by an external magnetic field, as the authors suggest for their own system. There are probably more examples I am not aware of. I have a hard time to see how the setup described by the authors would present a higher level of controllability and tunability than the ones described in these previous publications. I think the authors should mention these works (and similar I would have missed) during the introduction and highlight the interest of their own setup in comparison to these.

As the authors pointed out, their observation of various clustered phases (clustered, stripes and labyrinth) indicate that their system might present closer similarities to microscopic system and serves as a model system for more general questions on mermaid potentials. However, it is not completely clear to me from the reported observations if this is the case. Indeed, their work currently does not discuss the fundamental conditions in which the different phases with aggregates (clusters, stripes and labyrinths) are observed. Therefore, it is not completely clear for now how whether these transitions and their detailed properties are universal or would closely match microscopic systems. For instance, there is no thermal agitation in their experiment, as is observed in colloidal suspensions, and the scaling laws of the observed forces are not matching the electrostatic repulsion between protons.

Therefore, although I'm convinced that the community of researchers working on floating objects will be thrilled by this work, I am not quite sure how it will be received by the broader readership that is usually attributed to Nature Communications. But maybe the authors have further arguments I am unaware of, that would support the use of their experiment as a model for microscopic systems, as the authors mentioned using it as a motivation? Or maybe those communities, with which I'm not so familiar, also have some fundamental questions about mermaid potentials and need a model system easier to observe to answer them?

Besides those limited reservations on literature review and broadness of interest (for which I might be wrong, since I'm not an expert on microscopic systems with mermaid potential), I think this manuscript is based on very sound methodology, with the highest accuracy standards for reproducibility of experiments and numerical simulations, with a very clear and comprehensive presentation of the authors' current results. It certainly deserves to be published almost as is.

Here are also a few minor remarks on the manuscript:

-In Fig. 1, authors mention red, violet and blue curves. In the PDF I have, it is not clear which curve is 'red' or 'blue'. I have a dark green/turquoise color, and a fade orange/brown one. I think the description of these colors need to be updated (if I have the right file).

-In the caption of Fig. 1, authors list the regimes as "the locally attractive or Cheerios (red) regime, the mermaid (violet) regime, and the fully repulsive (blue) regime." And then say they are linked to increasing masses of " $m_1 = 0.080$

g, $m_2 = 0.092$ g, and $m_3 = 0.11$ g, respectively". If I understood correctly, the mass should be listed in the reverse order, as a heavier object would have a stronger capillary attraction, wouldn't it ?

-The authors mention line 119 "If $F_p(l = 0) > 0$ (i.e. locally repulsive), F_p can have either 0, or 2 roots for $l > 0$." Maybe it would be a good idea to highlight the discussed force roots (energy extrema) in Fig. 1 for more clarity ?

-In Fig. 2, I would highlight the experimental points depicted in subpanel (b) and (c) with a different border color or texture in panel (a) (otherwise, it is not so convenient to identify the exact point described by those snapshots).

-Fig. 3: are there objective criteria that the authors apply to distinguish clusters from stripes and labyrinths ? For instance, are there reference $\bar{g}(r)$ for those states in other systems that the authors could use for comparison ?

-Although the manuscript is written with extreme care, I noticed that the authors started some sentences with acronyms (e.g. line 42) or mathematical symbols (e.g. line 98), which I've been told is to avoid.

Reviewer #3 (Remarks on code availability):

The code is provided in a Github repository with an appropriate readme file. although I needed to install a few packages in order to run the python code, I got it up and running within an hour.

The authors couldn't have done better nor clearer to share this resource.

Authors' Response to Reviews of

Mermaid Cereal: Interactions and Pattern Formation in a Macroscopic Magnetocapillary SALR System

Alireza Hooshanginejad, Jack-William Barotta, Victoria Spradlin, Giuseppe Pucci, Robert Hunt, Daniel M. Harris

Nature Communications, NCOMMS-24-20586-T

RC: Reviewers' Comment, AR: Authors' Response, Manuscript Text

Reviewer 1

RC: *In this work, the authors used floating millimetric disks at a fluid interface, urged by a capillary attraction. The disks also have permanent magnets to introduce a competing repulsion effect. They study their pattern formation in equilibrium, where a short-range attraction and long-range repulsion interaction potential describe the pairwise energy landscape of two disks. They do experiments and simulations, demonstrating that the equilibrium pattern can be externally manipulated by applying a supplemental vertical magnetic force that enhances the effective capillary attraction.*

In my opinion, this is an outstanding work. The data obtained through experiments sounds technically well-developed and exhaustively analyzed and interpreted. Also, I consider the author's conclusions consistent with their findings. In this manuscript, one can learn the physical consequences of modifying common potentials (repulsion at the short range and attraction at the long-range). Ideal for teaching statistical physics. As the authors wrote: The current magnetocapillary SALR system provides an accessible playground for investigating analogous aspects of SALR systems across colloidal scales to subatomic scales where experiments are more challenging or expensive, and could also be used in pedagogy when introducing modern topics in self-assembly. This manuscript must be published.

AR: We thank the referee for their strong support and minor suggestions. In what follows we address each of their minor comments, which we believe have improved the manuscript. All changes made to the manuscript are indicated in blue.

RC: *However, I have a few questions that have to be addressed by the authors:*

1) In lines 196-198, they mention that fully repulsive disks exhibit a hexagonal crystal lattice. This surely occurs for large corrals, but they are only distorted hexagons here. Please comment on that.

AR: Indeed, the relatively small size of the experimental corral leads to defects in the hexagonal lattice. Experimentally this is true as the referee notes, and is also recovered in an analogous numerical simulation (Figure R1) where similar defects arise. For example, in the center of this configuration, one can clearly see a defect in the crystalline order with a pentagon of disks (rather than a hexagon of disks). To contrast to this more confined case, we simulated an experiment with double the corral radius, and hence quadruple the area. This required 228 disks (instead of 57). A much clearer hexagonal crystalline order is present, apart from localized regions near the boundary. In addition to these confinement effects, the experiment inevitably has minor variations in parameters such as the disk mass and magnetic moment that may also contribute to non-ideal lattice configurations.

Figure R1: Regularization of the lattice. For a simulation with fixed packing fraction of fully repulsive disks, as the domain size is expanded (weakening the effects of confinement on pattern formation) a more regular hexagonal lattice emerges. The experimental size (left) is compared to a domain that is four times larger in area (right).

Overall this is a valuable point raised, and we should be more precise in our wording of the particular section highlighted. We have now modified the mentioned section:

As shown in Fig. 2(b), fully repulsive disks with $a = 3$ mm, $m = 0.073$ g, and $M = 9$ A·cm⁻² ($\mathcal{M} = 2.1$ and $Bo = 1.52$) exhibit a hexagonal crystal lattice with defects at $\phi = 0.1425$. The hexagonal lattice has an inhomogeneous structure as a result of the finite confinement. For lesser confinement, the lattice structure is expected to become more regular.

RC: 2) *Is the order long- or short-ranged for the average radial distribution function RDF, $g(r)$, for the first column in Fig 3?*

AR: The order exists on a length scale set by the hexagonal lattice spacing for the particular packing fraction used. However, due to the periodicity of the pattern, additional peaks also occur at larger distances (much larger than the mermaid equilibrium spacing where $\bar{l} \gg 1$) suggesting long-range order, similar to that observed for a crystal. While it is difficult to definitively draw a line between long- and short-range, the RDFs do enable us to make useful comparisons about the relative extent of order between the various states. We have now added a statement to this effect:

In comparison to the low-packing fraction hexagonal lattice state, the intermediate packing fractions display a more short-range order. As the packing fraction is further increased, it is anticipated that a long-range order should be restored as a dense hexagonal lattice emerges, with particle-particle separation approximately $\bar{l} = 1$.

RC: 3) *In Methods, you must include details of the used simulation method.*

AR: Thank you for bringing up this point. We have added a new Methods subsection reading:

All simulations are carried out in MATLAB using the known physical parameters from the corresponding experiments as detailed in the figure captions. The ode45 solver (explicit Runge-Kutta (4,5) formula) in MATLAB is used to iterate the system of ordinary differential equations for both the confined and periodic domains. All simulations begin by randomly placing N disks across the domain, sampling from a uniform distribution. Resampling occurs in the case where two positions are sampled that would correspond to a physical overlap of two disks in experiment. For the case of circular confinement, the sampling region of the domain is restricted to a circle of radius $R - \ell_c$ so that no disks are placed on the meniscus of the container of radius R . Eq. 7 was simulated for a sufficiently long time such that the energy of the final configuration was constant for an extended period, ensuring that a minimal energy configuration was attained.

For the periodic domain, we initialize N particles inside a square domain of side length L . At each time step, we place an image square of disks to the left, right, above, below, and on all four of the diagonals, and compute the total net forces on the original particles. Due to large size of the domain relative to the extent of the competing interaction forces (magnetic/capillary), no additional images are needed to mimic a period boundary. As such, each individual particle responds to forces from $9N - 1$ particles (i.e. $N - 1$ particles in the original domain, and $8N$ image particles). No additional confinement potential is used in the periodic simulations.

For the purpose of computing the RDFs from the outputs of the simulations we set $\Delta r = \ell_{eq}$, which is a small but physically meaningful length scale in the problem.

We now reference this extended Methods in the main text:

The underdamped version of Eq. 7 (i.e. $4N$ equations) gives indistinguishable results for the equilibrium state, justifying the over-damped simplification for our particular exploration. Additional details on the simulation setup and implementation can be found in Methods.

Additionally, all commented code is provided in our digital repository¹ enabling full reproducibility of the results.

¹<https://github.com/harrislaboratory/MermaidCereal>

Authors' Response to Reviews of

Mermaid Cereal: Interactions and Pattern Formation in a Macroscopic Magnetocapillary SALR System

Alireza Hooshanginejad, Jack-William Barotta, Victoria Spradlin, Giuseppe Pucci, Robert Hunt, Daniel M. Harris

Nature Communications, NCOMMS-24-20586-T

RC: Reviewers' Comment, AR: Authors' Response, Manuscript Text

Reviewer 3

- RC:** *The authors describe an original experimental system, where floating objects embedded with a permanent magnets present various interaction regimes. In the most complex regime, the long range repulsion generated by the magnets and the short range attraction induced by the Cheerios effect compete to create a "mermaid" potential, similar to microscopic systems. The most original observation of the authors resides in the apparition of various self-assembled phases due to this mermaid potential, similar to what is observed in microscopic scales. The authors determined the conditions in which the mermaid potential is obtained with analytical equations, and also provide numerical simulations matching closely their experiment. The work of the authors is of very sound nature and they provide all required information to reproduce their experiments and simulations, including some open access version of their code, which I really appreciate. Their results are potentially interesting as they could open the way to mimic structures observed with microscopic systems.*
- AR:** We thank the referee for their support and suggestions. In what follows we address each of their specific comments, which we believe have improved the manuscript. All changes made to the manuscript are indicated in blue.
- RC:** *Regarding the literature review, I think that the authors overlooked a few of the most recent publications involving controlled self-assembly of floating objects. Here are some examples from a lab I am familiar with:
DOI: 10.1039/d0sm01251c
DOI: 10.1039/d1sm00447f
DOI 10.1140/epje/i2017-11599-y
The last one (Metzmacher, Poty et al., 2017) in particular also studies the self-assembly of floating objects with embedded permanent magnets, whose capillary interaction is governed by an external magnetic field, as the authors suggest for their own system. There are probably more examples I am not aware of. I have a hard time to see how the setup described by the authors would present a higher level of controllability and tunability than the ones described in these previous publications. I think the authors should mention these works (and similar I would have missed) during the introduction and highlight the interest of their own setup in comparison to these.*
- AR:** We greatly appreciate the reviewer bringing these more recent works to our attention, and fully agree that they should be included in our literature review. We have previously cited a number of the seminal works from this group on magnetocapillary phenomena in the penultimate paragraph of the introduction, however we regret that these additional references that focus on the *control* of capillary assembly were not originally included.

We have now included these references as well as others from the same group in the introduction:

Remote control and actuation of capillary assemblies is also a topic of recent interest, realized for instance through the external control of capillary interactions of deformable objects [1, 2], the application of time-dependent external fields to which the particles respond dynamically [3], or the introduction of new physical effects such as thermocapillarity [4].

We later reference the Metzmacher et al. paper [1] again when describing our experiments involving pattern control with an additional magnetic sheet:

While in our case the particles themselves do not deform under the additional applied field, other recent work has demonstrated more advanced control of capillary multipole interactions through magnetically responsive flexible floating objects [1].

While this paper also used embedded permanent magnets in their floating objects, the primary reason for doing so was to enable a clever non-contact method for deforming the floating objects via a Helmholtz coil arrangement. Although lateral particle-particle magnetic repulsion was noted, it was not characterized in any detail beyond observation.

The referee also mentions another recent work by Poty & Vandewalle [5] that describes an interesting mechanism by which an equilibrium spacing between two floating objects can be realized without the need for magnetic repulsion, in the case where the objects' heights do not vary during the interactions. While not as immediately relevant to the present study, we have now added a citation to this very interesting discovery in the introduction:

Capillary attraction between floating objects is a fundamental problem with relevance to applications involving self-assembly at fluid interfaces in natural systems [6, 7, 8, 9] and laboratory settings [10, 11, 5, 12].

Our current system shares most similarity with the earlier papers from the referenced group on the magnetocapillary interactions of small spherical beads (e.g. [13]), but we would like to emphasize some key distinguishing features here. In particular, they do not document long-range repulsion in their work, nor the accompanying gamut of pattern formation known for such systems, as we have clearly demonstrated here. There are a number of possible reasons why this may not have been a focus (including operating in a different parameter regime (i.e. very small Bo)), but we would prefer not to speculate and instead focus on the claims and observations. Secondly, these prior works rely on an external magnetic field provided by a DC-powered Helmholtz coil to induce the magnetic dipoles in the particles. While this does enable an additional level of control, it is more difficult to scale and inherently less accessible. None of our results rely upon externally powered equipment or precise alignment of external apparatuses, and the possible domain size is only set by the physical size of the fluid container alone (rather than also being restricted by the spatial details of a coil-generated applied magnetic field). These key distinctions are briefly described in the introduction:

To prevent complete particle aggregation under capillary attraction, prior works have introduced repulsive magnetic dipole-dipole interactions through the application of an external magnetic field [14, 13, 3, 15]. However, previous work at the macroscale has predominantly focused on short-range effects, with limited attention to the consequences of the relative long-range nature of the magnetic repulsion [13]. Furthermore, the region of approximately uniform magnetic field in the Helmholtz coil geometry used in the prior experiments sets practical limits on the number of particles and the size of

the domain that can be considered.

It is our hope that our system will ultimately complement the existing and ongoing work from this group, and are glad the referee brought these additional relevant works to our attention so that we could cite them appropriately.

RC: *As the authors pointed out, their observation of various clustered phases (clustered, stripes and labyrinth) indicate that their system might present closer similarities to microscopic system and serves as a model system for more general questions on mermaid potentials. However, it is not completely clear to me from the reported observations if this is the case. Indeed, their work currently does not discuss the fundamental conditions in which the different phases with aggregates (clusters, stripes and labyrinths) are observed. Therefore, it is not completely clear for now how whether these transitions and their detailed properties are universal or would closely match microscopic systems. For instance, there is no thermal agitation in their experiment, as is observed in colloidal suspensions, and the scaling laws of the observed forces are not matching the electrostatic repulsion between protons.*

AR: We thank the reviewer for this important comment. While it is indeed true that the exact functional form of a given potential will likely shift the onset of the various aggregates forming, the sequence of observed phase behaviors as a function of the control parameters (notably here the packing fraction) is still expected. For example, prior theoretical analysis has been conducted on systems with competing interactions wherein a potential is given as a sum of two terms: one attractive and one repulsive [16]. Notably, the functional form of the two terms is kept constant (modified Bessel function of the first kind) but they act over different length scales (set by a parameter in the argument of the Bessel function) and with different strengths (set by a multiplicative prefactor). In the analysis that follows, the authors find that this relatively generic system is able to exhibit clustering, striping, labyrinths, and eventually a dense triangular lattice at high packing fractions (See Fig. 7 [16]). In addition, while these simulations were *initialized* with some level of “temperature,” they are then cooled to zero (with “zero temperature” being equivalent to our non-Brownian system) to study the generic pattern formation.

Other analyses for competing interactions use entirely different functional forms (such as combinations of polynomials and exponentials) and predict the same set of states [17]. In fact, this latter work additionally discusses the role of thermal agitation in more detail and demonstrates that while the phase boundaries may shrink as a result, the general qualitative pattern formation behavior is largely unaffected (see Fig. 1 [17]). The only exception is at very high temperatures, where the thermal agitation energy exceeds the interaction potential, and the particles remain dispersed.

Thus while the precise mathematical forms of the competing interactions inevitably differ between physical systems (and models), the existing literature indicates that the gamut of emergent patterns formed is rather universal for SALR systems. We have added the following sentence to the introduction to help clarify this point for the reader:

Nevertheless, while the precise mathematical form of the competing interactions inevitably differ between physical systems, the gamut of emergent patterns formed appears rather universal [17, 16].

We do believe that including an effective thermal agitation in our macroscopic experimental system would increase the richness of the analogy in future work. A promising future direction would be to consider chaotic waves on the interface as a proxy for thermal agitation, as we have begun to explore in other work [18].

RC: *Therefore, although I’m convinced that the community of researchers working on floating objects will be thrilled by this work, I am not quite sure how it will be received by the broader readership that is usually*

attributed to Nature Communications. But maybe the authors have further arguments I am unaware of, that would support the use of their experiment as a model for microscopic systems, as the authors mentioned using it as a motivation? Or maybe those communities, with which I'm not so familiar, also have some fundamental questions about mermaid potentials and need a model system easier to observe to answer them?

AR: As evidenced through the numerous recent references (including many to SALR/mermaid potentials, specifically), self-assembly processes guided by understanding and control of pairwise interaction potentials represent an active area of research in condensed matter physics. The current paradigms for studying SALR systems, specifically, include microscopic experiments and computer simulations. Our macroscopic system represents a third *complimentary* approach, with its own unique benefits and drawbacks, in which to study this rich class of systems. As discussed in the recent Review Article on mermaid potentials in *Soft Matter* [19], microscale experiments are often difficult to quantitatively model due to the subtle (and often coupled) physics involved at these scales. In contrast, and as recognized by the reviewers, the physics of our experimental system is well validated in the present work. We have updated the manuscript to describe this recognized challenge:

However, direct comparison between experiment and theory is frequently challenging, due to the subtle and often coupled physics involved in microscale realizations [19].

As concrete future steps, there are a number of theoretical and numerical predictions that lack experimental validation due to the challenge and complexity of experiments at the microscale, such as the formation of spiral patterns in confinement [20] or additional pattern control over periodic substrates [21]. We hope that our system may be an accessible test bed to validate such predictions and motivate similar explorations at the microscale. As described in the prior point, while there may be subtle mathematical differences between systems, the emergent phenomena are relatively insensitive to such details. Additionally, the visual and interactive hands-on nature of our experiment may allow researchers to gain an increased physical intuition on these systems. We have updated the discussion to better describe the possible role of our system in the future:

The current magnetocapillary SALR system provides an accessible playground for investigating analogous aspects of SALR systems across colloidal scales to subatomic scales where experiments are more challenging or expensive. For instance, there are a number of recent theoretical and numerical predictions that lack experimental validation that may be good candidates for future work with our system, such as the formation of spiral patterns in confinement [20] or pattern formation over periodic substrates [21]. Additionally, the system could be fruitfully applied to pedagogy for introducing modern topics in self-assembly or allow researchers to develop a better physical intuition on analogous microscale systems.

While at this stage it is difficult to definitely prove this system will be uptaken by the community as a model system, this is certainly one of the goals of our work. Publication in *Nature Communications* would afford our work an additional visibility that will likely prove helpful to this goal moving forward, while also further promoting the richness of such capillary systems. Although of course only anecdotal, since sharing our work in the *APS Gallery of Soft Matter* and on *arXiv*, our group has been contacted by a number of researchers in the field of colloidal self-assembly with great interest and enthusiasm. One such researcher recently noted in email that “[y]our work on Mermaid Cereal and SALR potentials was a source of inspiration for our work.” Another individual shared recent theoretical predictions from their group with the hope that we might explore such predictions with our system and simultaneously navigate uncharted territory together in the future.

Perhaps more philosophically, the concept of “physical analogy” wherein there is a “partial similarity between the laws of one science and those of another which makes each of them illustrate the other” (J.C. Maxwell, 1855) has been fruitfully leveraged throughout the history of science with the hope that “the two systems of ideas leads to a knowledge of both, more profound than could be obtained by studying each system separately” (J.C. Maxwell, 1870). It is thus possible that observations at the microscale may in return extend our understanding of capillary assembly phenomena, by cooperative exploration of the scope and limitations of such an analogy.

Lastly, while some papers published in *Nature Communications* may appeal to broader audiences, and we certainly hope that ours will as well, this is not an explicit requirement of the journal (in contrast to others): the journal’s Aims only describe that “papers published by the journal represent important advances of significance to specialists within each field.”¹ The fact that the reviewer has definitively identified at least one active community of researchers that will find great interest in this work (i.e. community of researchers working on floating objects) would suggest that we have satisfied this criterion. Fundamental aspects of self-assembly have been a topic of focus for numerous recent articles in *Nature Communications* (e.g. [22, 23, 24]) and thus we believe it will attract similar readership.

RC: *Besides those limited reservations on literature review and broadness of interest (for which I might be wrong, since I’m not an expert on microscopic systems with mermaid potential), I think this manuscript is based on very sound methodology, with the highest accuracy standards for reproducibility of experiments and numerical simulations, with a very clear and comprehensive presentation of the authors’ current results. It certainly deserves to be published almost as is.*

AR: We are very pleased to hear that the reviewer finds the scientific content to be fully sound and reproducible, as this is always the primary goal of our shared work.

RC: *Here are also a few minor remarks on the manuscript:*

-In Fig. 1, authors mention red, violet and blue curves. In the PDF I have, it is not clear which curve is 'red' or 'blue'. I have a dark green/turquoise color, and a fade orange/brown one. I think the description of these colors need to be updated (if I have the right file).

AR: An excellent point, as we strayed too far from our intended color palette. We have now updated the colors to be more faithfully red and blue. Similar updates were made to the supplementary materials for clarity.

RC: *-In the caption of Fig. 1, authors list the regimes as "the locally attractive or Cheerios (red) regime, the mermaid (violet) regime, and the fully repulsive (blue) regime." And then say they are linked to increasing masses of "m1 = 0.080 g, m2 = 0.092 g, and m3 = 0.11 g, respectively". If I understood correctly, the mass should be listed in the reverse order, as a heavier object would have a stronger capillary attraction, wouldn't it ?*

AR: The reviewer is certainly correct. This has now been updated:

The three characteristic regimes are shown here (corresponding to the sequel snapshots in the inset): the locally attractive or Cheerios (red) regime, the mermaid (violet) regime, and the fully repulsive (blue) regime. For all three cases $a = 3$ mm and $M = 9$ A·cm² with decreasing masses of $m_1 = 0.11$ g, $m_2 = 0.092$ g, and $m_3 = 0.080$ g, respectively.

RC: *-The authors mention line 119 "If $F_p(l = 0) > 0$ (i.e. locally repulsive), F_p can have either 0, or 2 roots for*

¹<https://www.nature.com/ncomms/aims>

l > 0." Maybe it would be a good idea to highlight the discussed force roots (energy extrema) in Fig. 1 for more clarity ?

AR: This is a nice idea, as it helps us more clearly connect these curves to our theoretical analysis. Markers at the energy extrema have been added, and the caption updated:

The circle (○) and diamond (◇) markers indicate the locations of energy extrema for stable and unstable equilibria when $l > 0$, respectively.

RC: *-In Fig. 2, I would highlight the experimental points depicted in subpanel (b) and (c) with a different border color or texture in panel (a) (otherwise, it is not so convenient to identify the exact point described by those snapshots).*

AR: We have now accomplished this as suggested by adding a distinctive white border to the points in (a) that correspond to (b) and (c), and updated the caption accordingly:

(b) The equilibrium configuration in experiment for fully repulsive disks with $\mathcal{M} = 2.1$, $\text{Bo} = 1.52$, and $\phi = 0.1425$ ($a = 3$ mm, $m = 0.073$ g, $M = 9$ A·cm⁻², $R = 6$ cm, $N = 57$), also marked with the white border right triangle in the phase diagram. (c) The equilibrium configuration in the experiments for mermaid disks with $\mathcal{M} = 1.22$, $\text{Bo} = 1.52$, and $\phi = 0.1425$ ($a = 3$ mm, $m = 0.092$ g, $M = 9$ A·cm⁻², $R = 6$ cm, $N = 57$), also marked with the white border circle in the phase diagram.

RC: *-Fig. 3: are there objective criteria that the authors apply to distinguish clusters from stripes and labyrinths ? For instance, are there reference $\bar{g}(r)$ for those states in other systems that the authors could use for comparison ?*

AR: This is a great point to raise, and one we struggled with ourselves. To the best of our knowledge, there does not exist a universal objective criterion for identifying the various regimes of pattern formation (lattice, cluster, stripe, etc.). In considering the wider literature on pattern formation, there does not even seem to exist a standard set of nomenclature for such states, with other terms such as droplets, bubbles, and void rich lattices appearing, for instance. One proposition for analysis of the various patterns in SALR systems is provided in [16] which relies on signatures in the radial distribution function, $g(r)$, to help distinguish the states. This is the approach we elected to use as well and prompted the computation of such statistics. While this approach is quite useful for identifying differences between lattices, cluster, and stripes/labyrinths given differing fundamental features, we certainly acknowledge that there nevertheless exists some subjectivity in the labeling, particularly in the difference between stripes and labyrinths.

For our specific case, given our system's tendency to form stripes of single particle width, we took the approach of defining the transition from lattice to clusters when the dominant peak occurred at \bar{l} , and the appearance of integer multiple peaks of \bar{l} to indicate the transition to stripes, as we describe in the main text. We have edited the relevant section in an attempt to be more clear and transparent:

To characterize the system's phase under the mermaid potential, we average $g(r)$ over 25 independent simulations for each ϕ in a square domain of 17 cm \times 17 cm with periodic boundary conditions. Figure 3(c) shows the average RDF $\bar{g}(r)$ as a function of the distance from the disks normalized by the equilibrium distance between two isolated disks, namely $\bar{l} = (l + 2a)/(l_{\text{eq}} + 2a)$. As indicated in Fig. 3(c), $\bar{g}(r)$ is small at $\phi = 0.05$ for $\bar{l} < 2$, with a maximum at $\bar{l} \approx 3.1$ which is the characteristic spacing in the hexagonal lattice formation for this packing fraction. For $\phi = 0.1$, the strongest peak emerges at $\bar{l} \approx 1$ indicative of the magnetocapillary mermaid stable equilibrium state, which we identify as

the transition to the cluster phase. With increasing ϕ , the clusters coalesce into longer chains of disks, hence, additional peaks appear at integer multiples of $\bar{l} \approx 1$ indicative of the striped phase. The stripe and labyrinth phases exhibit similar RDFs [16], although the labyrinth phase has visual signatures of a more globally connected network of stripes.

As more peaks emerge, the gradual transition to labyrinths occur (wherein patterns look similar to stripes with perhaps more curvature in the stripes and more overall connectedness). To this end, potentially exploiting metrics from topology could prove fruitful, but remains speculative and room for future work. We note this idea in the manuscript:

These phases undergo a topological transition and could potentially be distinguished by alternative metrics, such as the corresponding Betti numbers or average pore size.

In fact, some authors make no attempt to distinguish between these phases, and simply refer to a single stripe phase with no mention of labyrinths [17].

RC: *-Although the manuscript is written with extreme care, I noticed that the authors started some sentences with acronyms (e.g. line 42) or mathematical symbols (e.g. line 98), which I've been told is to avoid.*

AR: These have now been corrected throughout.

RC: *Reviewer 3 (Remarks on code availability):*

The code is provided in a Github repository with an appropriate readme file. although I needed to install a few packages in order to run the python code, I got it up and running within an hour.

The authors couldn't have done better nor clearer to share this resource.

References

- [1] Jean Metzmacher, Martin Poty, Geoffroy Lumay, and Nicolas Vandewalle. Self-assembly of smart mesoscopic objects. *The European Physical Journal E*, 40:1–6, 2017.
- [2] Nicolas Vandewalle, Martin Poty, Nathan Vanesse, Jérémie Caprasse, Thomas Defize, and Christine Jérôme. Switchable self-assembled capillary structures. *Soft Matter*, 16(45):10320–10325, 2020.
- [3] G. Grosjean, M. Hubert, and N. Vandewalle. Magnetocapillary self-assemblies: Locomotion and micromanipulation along a liquid interface. *Advances in Colloid and Interface Science*, 255:84–93, 2018.
- [4] Ylona Collard, Franco N Piñan Basualdo, Aude Bolopion, Michaël Gauthier, Pierre Lambert, and Nicolas Vandewalle. Controlled transitions between metastable states of 2d magnetocapillary crystals. *Scientific Reports*, 12(1):16027, 2022.
- [5] Martin Poty and Nicolas Vandewalle. Equilibrium distances for the capillary interaction between floating objects. *Soft Matter*, 17(28):6718–6727, 2021.
- [6] D.L. Hu and J.W.M. Bush. Meniscus-climbing insects. *Nature*, 437(7059):733–736, 2005.

- [7] J.W.M. Bush, D.L. Hu, and M. Prakash. *The Integument of Water-walking Arthropods: Form and Function*, volume 34, pages 117–192. Academic Press, 2007.
- [8] J. Voise, M. Schindler, J. Casas, and E. Raphaël. Capillary-based static self-assembly in higher organisms. *Journal of The Royal Society Interface*, 8(62):1357–1366, 2023/08/15 2011.
- [9] P. Peruzzo, A. Defina, H.M. Nepf, and R. Stocker. Capillary interception of floating particles by surface-piercing vegetation. *Physical Review Letters*, 111(16):164501–, 10 2013.
- [10] N. Bowden, A. Terfort, J. Carbeck, and G.M. Whitesides. Self-assembly of mesoscale objects into ordered two-dimensional arrays. *Science*, 276(5310):233–235, 2023/09/07 1997.
- [11] I.B. Liu, N. Sharifi-Mood, and K.J. Stebe. Capillary assembly of colloids: Interactions on planar and curved interfaces. *Annual Review of Condensed Matter Physics*, 9(1):283–305, 2023/09/07 2018.
- [12] C. Zeng, M.W. Faaborg, A. Sherif, M.J. Falk, R. Hajian, M. Xiao, K. Hartig, Y. Bar-Sinai, M.P. Brenner, and V.N. Manoharan. 3d-printed machines that manipulate microscopic objects using capillary forces. *Nature*, pages 1–6, 2022.
- [13] N. Vandewalle, L. Clermont, D. Terwagne, S. Dorbolo, E. Mersch, and G. Lumay. Symmetry breaking in a few-body system with magnetocapillary interactions. *Physical Review E*, 85(4):041402–, 04 2012.
- [14] M. Golosovsky, Y. Saado, and D. Davidov. Energy and symmetry of self-assembled two-dimensional dipole clusters in magnetic confinement. *Physical Review E*, 65(6):061405–, 06 2002.
- [15] G. Lagubeau, G. Grosjean, A. Darras, G. Lumay, M. Hubert, and N. Vandewalle. Statics and dynamics of magnetocapillary bonds. *Physical Review E*, 93(5):053117–, 05 2016.
- [16] H.J. Zhao, V.R. Misko, and F.M. Peeters. Analysis of pattern formation in systems with competing range interactions. *New Journal of Physics*, 14(6):063032, 2012.
- [17] Alessandra Imperio, Luciano Reatto, and Stefano Zapperi. Rheology of colloidal microphases in a model with competing interactions. *Physical Review E*, 78(2):021402, 2008.
- [18] S.J. Thomson, J-W. Barotta, and D.M. Harris. Nonequilibrium capillary self-assembly, 2023.
- [19] C.P. Royall. Hunting mermaids in real space: known knowns, known unknowns and unknown unknowns. *Soft Matter*, 14(20):4020–4028, 2018.
- [20] J Pękałski, E Bildanau, and Alina Ciach. Self-assembly of spiral patterns in confined systems with competing interactions. *Soft Matter*, 15(38):7715–7721, 2019.
- [21] C Reichhardt and CJO Reichhardt. Peak effect and dynamics of stripe and pattern forming systems on a periodic one dimensional substrate. *arXiv preprint arXiv:2402.18539*, 2024.
- [22] David Doan, John Kulikowski, and X Wendy Gu. Direct observation of phase transitions in truncated tetrahedral microparticles under quasi-2d confinement. *Nature Communications*, 15(1):1954, 2024.
- [23] Tianran Zhang, Dengping Lyu, Wei Xu, Xuan Feng, Ran Ni, and Yufeng Wang. Janus particles with tunable patch symmetry and their assembly into chiral colloidal clusters. *Nature Communications*, 14(1):8494, 2023.
- [24] H Dehne, A Reitenbach, and AR Bausch. Reversible and spatiotemporal control of colloidal structure formation. *Nature Communications*, 12(1):6811, 2021.

REVIEWERS' COMMENTS

Reviewer #3 (Remarks to the Author):

I thank the authors for their careful considerations of each and every of my comments, along a very detailed and justified answer. I'm glad to say that I learned more about SALR systems via this answer, and appreciate that the authors incorporated a summary of these teachings in the main text. I'm now convinced that the current work is of broader interest, even though it is not a formal requirement of the journal, as the authors pointed out. My apologies if I overstepped my role in that regard and deviated from the editorial guidelines in my previous review.

The authors have fully and convincingly answered all remarks I had. I would just like to point out that I think panel b) of Fig. 2 is related to the white border triangle and panel c) to the white border circle, i.e. reversed from the current caption (if I understood correctly). Despite this minor typo, I now fully support the publication of this manuscript in Nature Communications, and congratulate the authors for such a sound piece of work. I hope they continue to receive the attention they deserve from all the communities involved.

Reviewer #3 (Remarks on code availability):

As before, the code works perfectly and is shared in the best possible way.

Authors' Response to Reviews of

Interactions and Pattern Formation in a Macroscopic Magneto-capillary SALR System of Mermaid Cereal

Alireza Hooshanginejad, Jack-William Barotta, Victoria Spradlin, Giuseppe Pucci, Robert Hunt, Daniel M. Harris

Nature Communications, NCOMMS-24-20586-T

RC: *Reviewers' Comment*, AR: Authors' Response, Manuscript Text

Reviewer 3 (Second Round)

RC: *I thank the authors for their careful considerations of each and every of my comments, along a very detailed and justified answer. I'm glad to say that I learned more about SALR systems via this answer, and appreciate that the authors incorporated a summary of these teachings in the main text. I'm now convinced that the current work is of broader interest, even though it is not a formal requirement of the journal, as the authors pointed out. My apologies if I overstepped my role in that regard and deviated from the editorial guidelines in my previous review.*

AR: We thank the referee for their support, and believe the manuscript has improved as a result of their feedback.

RC: *The authors have fully and convincingly answered all remarks I had. I would just like to point out that I think panel b) of Fig. 2 is related to the white border triangle and panel c) to the white border circle, i.e. reversed from the current caption (if I understood correctly). Despite this minor typo, I now fully support the publication of this manuscript in Nature Communications, and congratulate the authors for such a sound piece of work. I hope they continue to receive the attention they deserve from all the communities involved.*

AR: This typo has now been corrected. We thank the referee again for their time and thoughtful feedback.

RC: *Reviewer 3 (Remarks on code availability):*

As before, the code works perfectly and is shared in the best possible way.